# Sugar-sweetened beverage consumption from 1998–2017: Findings from the health behaviour in school-aged children/school health research network in Wales

**Kelly Morgan**[1]*, **Emily Lowthian**[1], **Jemma Hawkins**[1], **Britt Hallingberg**[2], **Manal Alhumud**[3], **Chris Roberts**[4], **Simon Murphy**[1], **Graham Moore**[1]

**1** Centre for Development, Evaluation, Complexity and Implementation in Public Health Improvement (DECIPHer), School of Social Sciences, Cardiff University, Cardiff, United Kingdom, **2** Cardiff School of Sport and Health Sciences, Cardiff Metropolitan University, Cardiff, United Kingdom, **3** Applied Medical Sciences, Community Health Sciences, King Saudi University, Riyadh, Saudi Arabia, **4** Knowledge and Analytical Services, Welsh Government, Cardiff, United Kingdom

* Morgank22@cardiff.ac.uk

**Data Availability Statement:** The HBSC Data Management Centre coordinates the work with the international data file and the trend data, and is the

## Abstract

To date no study has examined time trends in adolescent consumption of sugar-sweetened beverages and energy drinks, or modelled change in inequalities over time. The present study aimed to fill this gap by identifying historical trends among secondary school students in Wales, United Kingdom. The present study includes 11–16 year olds who completed the Health Behaviour in School-aged Children (HBSC) survey and the Welsh School Health Research Network (SHRN) survey between 1998 to 2017. Multinomial regression models were employed alongside tests for interaction effects. A total of 176,094 student responses were assessed. From 1998 to 2017, the prevalence of daily sugar-sweetened beverage consumption decreased (57% to 18%) while weekly consumption has remained constant since 2006 (49% to 52%). From 2013 to 2017, daily consumption of energy drinks remained stable (6%) while weekly consumption reports steadily decreased (23% to 15%). Boys, older children and those from a low socioeconomic group reported higher consumption rates of sugar-sweetened beverages and energy drinks. Consumption according to socioeconomic group was the only characteristic to show a statistically significant change over time, revealing a widening disparity between sugar-sweetened beverage consumption rates of those from low and high socioeconomic groups. Findings indicate a positive shift in overall consumption rates of both sugar-sweetened beverages and energy drinks. Adolescents from a low socioeconomic group however were consistently shown to report unfavourable sugar-sweetened beverages consumption when compared to peers from high socioeconomic group. Given the established longer term impacts of sugar-sweetened beverage and energy drink consumption on adolescent health outcomes, urgent policy action is required to reduce overall consumption rates, with close attention to equity of impact throughout policy design and evaluation plans.

Data Bank for the HBSC-study. The centre distributes data in accordance with the HBSC data access policy. Information on data access and materials can be located at https://www.uib.no/en/hbscdata Data from the Student Health and Wellbeing Survey are available upon reasonable request and abidance with the School Health Research Network's Data Use Protocol. Further information is available from shrn@cardiff.ac.uk.

**Funding:** This work was funded by a Health and Care Research Wales Health Fellowship Award [grant number HF-16-1164 to K.M.]. The Centre for Development, Evaluation, Complexity and Implementation in Public Health Improvement (DECIPHer) is funded by Welsh Government through Health and Care Research Wales. This work was supported also by the Public Health Division, Welsh Government with the support of The Centre for the Development and Evaluation of Complex Interventions for Public Health Improvement (DECIPHer), a UKCRC Public Health Research Centre of Excellence. Joint funding [grant number MR/K0232331/1] from the British Heart Foundation, Cancer Research UK, Economic and Social Research Council, Medical Research Council, the Welsh Government and the Wellcome Trust, under the auspices of the UK Clinical Research Collaboration, is gratefully acknowledged. The funders had no role in study design, data collection and analysis, decision to publish, or preparation of the manuscript.

**Competing interests:** The authors have declared that no competing interests exist.

## Introduction

Consumption of sugar-sweetened beverages (SSBs), including energy drinks (EDs) represents a significant public health problem, with consumption rates linked to an increased health risk of type II diabetes [1] and dental erosion [2]. Soft drinks contribute an estimated 40% of sugar intake among adolescents [3], of which EDs make up an increasing proportion [4]. In 2017, one study found that 95% of EDs would receive a 'red' (high) label for sugars per serving [5]. This poses a concern as dietary patterns track from adolescence into adulthood [6], and this period represents a crucial phase in the life-course for the development of various diseases [7].

SSBs, including EDs are widely available and promoted. Marketing strategies have actively targeted certain communities, for example, using outdoor advertisements within deprived areas, and increased television exposure among young people within minority ethnic and low-income communities [8]. It is estimated that 1 billion litres of soft drinks are produced globally each year [9] with the soft drinks industry contributing £11 billion to UK economic growth [10]. The EDs market is estimated to be worth over £2 billion in the UK [11] and $50 billion globally, with a projected annual growth rate of 3.5% between 2015 and 2020 [12]. Concerns around ED consumption primarily relate to the high caffeine content, however some large ED cans may contain up to 21 teaspoons of sugar [13], over three times the daily recommendation [14].

While calls for a reduction in SSB consumption date back to 1942 in the United States (US), only in recent years have EDs received increasing attention from policy-makers and health experts [5, 15]. Some countries have banned sales of EDs and others have introduced sales and labelling regulations [11]. At present there remains no UK-wide legislation relating specifically to EDs. In 2018, most major UK supermarket chains enforced a ban on ED sales to under 16s [16] and 2019 saw a ban on ED sales to under 16s in all NHS sites in Scotland [17]. An ED ban for under 16s was proposed by the UK government in 2018, but has not yet been executed [18]. Instead, devolution in each UK nation has resulted in a number of consultations. The Scottish Government recently closed a consultation on ending the sale of EDs to under 16's [19]. In December 2019, the Welsh Government set out plans to ban sales of EDs to all children and young people by 2030 as part of a nationwide strategy [20]. Thus, present UK sales remain unregulated with a voluntary code of practice to avoid deliberate marketing of products to under 16's [11], and stakeholders and health experts across all levels continue to call on industry and government to introduce a ban on such sales.

To drive product reformulation and reduce sugar content, a two-tiered Soft Drinks Industry Levy (SDIL) was introduced in the UK in 2018 [21]. Including EDs, the policy concerns the production and importation of SSBs and aims to incentivise manufacturers to lower sugar content though the lowering of tax rates, referred to as the 'Sugar Tax'. Recent UK findings show this is having a favourable impact on the sugar content in drinks [22].

Trends in SSB consumption have been documented widely across young people in the US [23] and recently Denmark [24], revealing declines in daily consumption over time. A 2017 report on daily SSB consumption rates, involving 32 European countries also noted a decline over time, yet results were limited to two time-points, 2002 and 2014, with no ED data [25]. As such, no study to date has examined periodic time trends in SSB and ED consumption among UK nations in the lead up to the introduction of the SDIL.

Little attention has been paid specifically to EDs, with only one US-based study to date exploring adolescent ED consumption trends with no time trend reports by demographic characteristics [26].

The aim of this study is to examine the consumption frequency of SSBs and EDs among 11–16 year olds over time. With use of national data collected between 1998 and 2017, we

examined overall consumption and reports among sociodemographic subgroups. Data available for SSB consumption spanned a 20-year period (1998–2017) and ED consumption across five years (2013–2017).

## Materials and methods

### Study sample

Student self-report data from the Health Behaviour in School-aged Children (HBSC) survey and the School Health Research Network (SHRN) surveys in Wales, from 1998 to 2017 were used. Surveys were conducted approximately every two years from 1998 to 2017. Data are appended over the years to create a repeated cross-sectional dataset as in previous studies [27]; Data on SSBs were available from 1998–2017, and EDs from 2013–2017. The HBSC survey, a collaborative cross-national survey, is administered every four years and currently involves 50 countries and regions across Europe and North America. The SHRN survey is administered every two years and is based on the HBSC survey allowing integration of the two surveys every four years. Over-time the SHRN survey sample size has grown due to the increasing number of schools in Wales agreeing to conduct the survey. Details on study sampling strategies and procedures can be accessed elsewhere [28, 29].

### Sociodemographic characteristics

Gender (response options: 'Boy' and 'Girl') and school year were reported in all survey years. School year and corresponding age groups were: Year 7 (age 11–12), Year 8 (age 12–13), Year 9 (age 13–14), Year 10 (age 14–15) and Year 11 (age 15–16). An indicator of socioeconomic status (SES) was available from 2002, using the Family Affluence Score (FAS) [30, 31] which comprised measures of: car and computer ownership, bedroom occupancy and family holidays. From 2013 onwards, two additional measures (dishwasher and bathroom ownership) were included [32]. Scores for each of the four/six survey items were summed for a total score, whereby a higher score indicated greater affluence. This score was split at the median in each survey year to achieve 'low' and 'high' SES.

### Definitions of outcome variables

**SSB consumption.** A question on SSB consumption, included in every survey year, asked; 'How many times a week do you usually drink Coke or other soft drinks that contain sugar?' (response options: 'Never', 'Less than weekly', 'weekly', '2–4 times a week', '5–6 times a week', 'Daily' and 'More than once a day'). In the first two survey years, 'Never' and 'Less than weekly' formed one category. For each survey, responses were recoded into a three-category variable indicating: 'Never or less than weekly' (includes 'Never' and 'Less than weekly'), 'Weekly' (includes 'Weekly/Once a week', '2–4 times a week' and '5–6 times a week') and 'Daily or more' (includes 'Daily' and 'More than one a day').

**ED consumption.** A question on ED consumption, included in 2013, 2015 and 2017, asked 'How many times a week do you usually drink energy drinks (such as Red Bull, Monster, and Rockstar)?' (Response options: 'never', 'less than once a week', 'once a week', '2–4 days a week', '5–6 days a week', 'once a day, every day' and 'every day, more than once'). Responses were recoded to form a three-category variable; 'Never or less than weekly' (included 'Never' and 'Less than weekly'), 'Weekly' (included 'Weekly/once a week', '2–4 times a week' and '5–6 times a week') and 'Daily or more' (included 'Daily' and 'More than one a day').

**Inclusion and exclusion criteria.** For SSB data, 1.6% (n = 2,840) were missing. Following introduction in 2013, missing data for ED questions ranged between 0.2% and 2.2%. Analyses

focused on students in Years 7 to 11 (i.e. aged 11–16). In the years 1998–2002 and 2006, Years 8 and 10 were not available at the time of analysis. Hence as a sensitivity analysis, analyses were conducted with only Year 7, 9 and 11. As trends did not differ with either method, data for all year groups were retained. In 2017, gender included an additional response category ('prefer not to say'). As there was only one year of data on this group, the students selecting this response were set as missing (2%; n = 2,261).

**Ethical approval and consent to participate.** Schools signed and returned a commitment form to participate in the HBSC study; parents were sent information sheets and had the option of withdrawing their child from the study. Before the survey, participants were assured of anonymity and confidentiality and asked to provide written active assent. All students had the opportunity to withdraw from data collection at any time. The survey was approved by Cardiff University Social Sciences Research ethics committee.

**Statistical analysis.** Statistical analysis was conducted using STATA 15. Descriptive data are presented as frequencies and percentages. Analyses of SSB and ED consumption over time were performed using multinomial regression models with time (variable year) included as a covariate to measure the effects of time on consumption. Multinomial logistic regression models examined associations between sociodemographic characteristics and the three-category variables for SSB and ED consumption (reference category—'Daily consumption or more'). Models were first tested for SSBs and EDs separately using the predictors of gender, school year, SES, and survey year. Coefficients are reported as relative risk ratios (RRR's).

Interaction effects with the variable 'year' and the characteristics of school year, gender and SES were also investigated to estimate change over time. All variables were mean-centred where applicable, i.e. not binary indicators, to limit multi-collinearity in analysis. Models were performed separately to test for change over time among characteristics of interest (e.g. gender). Interactions were estimated using multinomial regression and predictive margins, and graphed using these estimates. Models were conducted using complete case analysis; other options were considered–see S1 File.

## Results

The total sample comprised 176,094 student responses (S1 Table provides a breakdown across each survey wave). Sample demographics (Table 1) were evenly split in terms of gender and SES. The largest school year group was Year 7 and the smallest Year 10. For SSBs, approximately one in two students reported weekly consumption (52%) overall, whereas just over a quarter never consume them (27%), and around one fifth reported daily or greater consumption (22%). For EDs, most students reported never consuming them, or consumption less than weekly (77%), whereas approximately 1 in 6 (17%) reported weekly consumption. Only 6% reported ED consumption daily or more. Cross-tabulations (S2 Table) showed a relationship between SSB and ED consumption, with daily SSB consumption being largely related to daily ED consumption, and vice-versa for never consumption of SSBs and EDs ($\chi^2$ = 24000, $p<0.05$, n = 140,470).

### Time trends

Time trend analysis results are shown in Figs 1 and 2. The proportion of students consuming SSBs daily decreased steeply from 2000 to 2006, appearing to plateau from 2009, dropping from 57% to 18% across the time series in 2017. Similarly, the proportion reporting never or less than weekly consumption increased four-fold from 7% in 1998 to 29% in 2017. Weekly SSB consumption increased steadily since 2000 and has remained constant since 2006 at 49% to 52% in 2017. More detailed analysis showed that the 'Once a week' and '2–4 times' a week were mostly attributable to the increase (See S13 Table). Regression analyses indicate that

**Table 1. Sample characteristics of the study participants (11–16 years) between 1998 and 2017 (n = 176,094).**

| Variable | N | | Total (n, %) | Missing (n, %) |
|---|---|---|---|---|
| **Gender** | *173 957* | | | 2337 (1%) |
| | | *Boy* | 85 919 (49%) | |
| | | *Girl* | 88 038 (51%) | |
| **School year** | *176 094* | | | 0 (0%) |
| | | *Year 7* | 40 358 (23%) | |
| | | *Year 8* | 34 467 (20%) | |
| | | *Year 9* | 39 200 (22%) | |
| | | *Year 10* | 30 570 (17%) | |
| | | *Year 11* | 31 499 (18%) | |
| **Socioeconomic Status** | *161 779* | | | 14315 (8%) |
| | | *Low* | 78 880 (49%) | |
| | | *High* | 82 899 (51%) | |
| **Sugary drink use** | *173 254* | | | 2,840 (2%) |
| | | *Never, or less than weekly* | 46 257 (27%) | |
| | | *Weekly use* | 89 228 (52%) | |
| | | *Daily use or more* | 37 769 (22%) | |
| **Energy drink use** | *141 154* | | | 34,940 (20%) |
| | | *Never, or less than weekly* | 109 208 (78%) | |
| | | *Weekly use* | 23 937 (17%) | |
| | | *Daily use or more* | 8 009 (6%) | |

relative to the highest consumption category (i.e. daily consumption), consumption of SSBs never or less than weekly increased significantly over time (*RRR* 1.08, $p<0.05$, CI 1.08–1.09), while weekly consumption also increased compared to daily consumption (*RRR* 1.05, $p<0.05$, CI 1.04–1.05). Hence, findings indicate an overall trend toward declining SSB consumption over time, indicated by increasing movement of the population toward lower consumption categories.

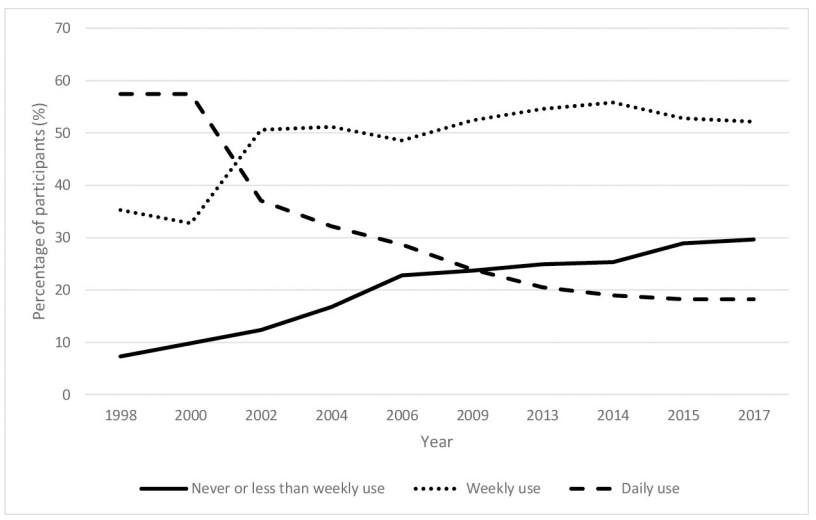

**Fig 1. Reported SSB consumption between 1998 and 2017.**

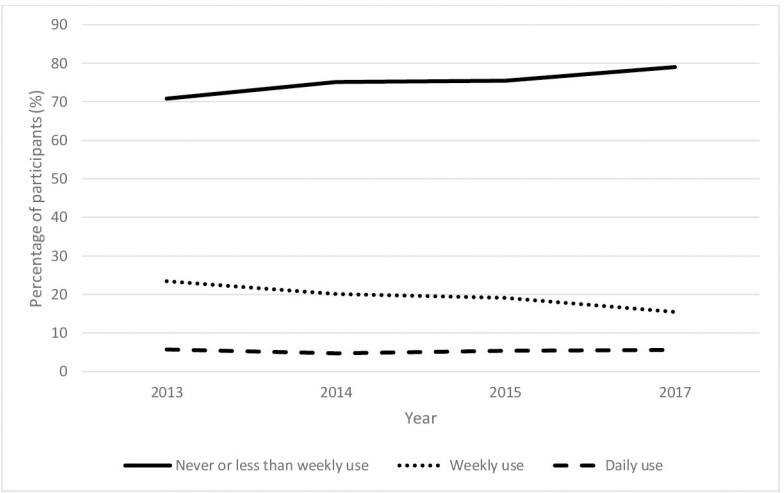

**Fig 2. Reported ED consumption between 2013 and 2017.**

Since 2013, daily ED consumption has remained stable at 6%. Weekly ED consumption has steadily decreased from 23% in 2013 to 15% in 2017. Conversely, a steady increase in reports of never or less than weekly consumption is estimated, 71% in 2013 vs. 79% in 2017. Upon further inspection, this steady increase reflects the increase in reports of 'Never' consumption over the years (see S15 Table). Regression analyses indicated that relative to daily consumption, reports of never or less than weekly remained unchanged over time relative to daily consumption (*RRR* 1.00, $p<0.05$, CI 0.98–1.02) although weekly consumption decreased over time relative to daily consumption (*RRR* 0.89, $p<0.05$, CI 0.87–0.91); note that when year was treated as categorical, RRR's showed an increase for never and less than weekly consumption (2015 *RRR* = 1.05, 2017 *RRR* = 1.03), and a decrease for weekly consumption (2015 *RRR* = 0.92, 2017 *RRR* = 0.68). Hence, while very regular consumption remains stable, the proportion of adolescents consuming EDs has fallen.

## SSB consumption and demographics

**Gender.** Against the reference category of daily consumption, girls were more likely to consume SSBs never or less than weekly compared to boys (*RRR* 1.75, $p<0.05$, CI 1.70–1.80). Boys were substantially less likely than girls to report never or less than weekly consumption (22% vs 31%) and slightly more likely to report daily consumption (24% vs 20%; S2 Table).

**Socioeconomic status.** Lower SES groups were less likely to report never or less than weekly consumption (*RRR* 0.68, $p<0.05$, CI 0.66–0.70) and weekly consumption (*RRR* 0.78, $p<0.05$, CI 0.76–0.80) when compared to daily consumption. Lower SES groups were more likely to be in the daily consumption group over-time compared to high SES groups (e.g. 21% vs 16% in 2017; S4 Table).

**School year.** Older students were less likely to consume SSBs never or less than weekly, compared to daily consumption; Year 9s and 10s were the least likely to report SSB consumption as never or less then weekly (*RRR* 0.70, $p<0.05$, CI 0.67–0.73/0.74). Year 8s had the highest likelihood of consuming SSB's never or less than weekly (*RRR* 0.86, $p<0.05$, CI 0.82–0.90). For weekly SSB consumption, a similar pattern was observed but Year 10's and 11's had the highest consumption risk (*RRR* 0.79, $p<0.05$, CI 0.76–0.83) and Year 8's had the lowest (*RRR* 0.95, $p<0.05$, CI 0.91–0.99). For the full model, see Table 2. Over time, 21% of Year 7's consumed SSBs daily, compared to 24% of Year 9's, and 25% of Year 11's (S6 to S10 Tables).

**Table 2. Multinomial regression of SSB consumption and sociodemographic characteristics with daily use as the reference category (n = 157,564).**

| | RRR | Std. Err | $p$ | Confidence Intervals | |
| --- | --- | --- | --- | --- | --- |
| | | | | Upper bound | Lower Bound |
| **Daily use** *(base outcome)* | | | | | |
| **Never or less than weekly use** | | | | | |
| **Gender** | | | | | |
| Girl | 1.75 | 0.03 | <0.001 | 1.70 | 1.80 |
| **School Year** | | | | | |
| Year 8 | 0.86 | 0.02 | <0.001 | 0.82 | 0.90 |
| Year 9 | 0.70 | 0.02 | <0.001 | 0.67 | 0.73 |
| Year 10 | 0.70 | 0.02 | <0.001 | 0.67 | 0.74 |
| Year 11 | 0.72 | 0.02 | <0.001 | 0.68 | 0.75 |
| **Socioeconomic Status** | | | | | |
| Low | 0.68 | 0.01 | <0.001 | 0.66 | 0.70 |
| **Year** | 1.08 | 0.00 | <0.001 | 1.08 | 1.09 |
| **Weekly use** | | | | | |
| **Gender** | | | | | |
| Girl | 1.11 | 0.01 | <0.001 | 1.08 | 1.14 |
| **School Year** | | | | | |
| Year 8 | 0.95 | 0.02 | 0.02 | 0.91 | 0.99 |
| Year 9 | 0.81 | 0.02 | <0.001 | 0.77 | 0.84 |
| Year 10 | 0.79 | 0.02 | <0.001 | 0.76 | 0.83 |
| Year 11 | 0.79 | 0.02 | <0.001 | 0.76 | 0.83 |
| **Socioeconomic Status** | | | | | |
| Low | 0.78 | 0.01 | <0.001 | 0.76 | 0.80 |
| **Year** | 1.05 | 0.00 | <0.001 | 1.04 | 1.05 |

## ED consumption and demographics

**Gender.** When comparing ED consumption to daily consumption, girls were more likely to report consumption as never or less than weekly compared to boys (*RRR* 1.85, *p*<0.05, CI 1.77–1.95), however this was not apparent for weekly consumption reports (*RRR* 1.00, *p* = 0.91, CI 0.94–1.05) (Table 3). Over time, 7% of boys consumed EDs daily compared to 4% of girls; with little change across years (see S3 Table).

**Socioeconomic status.** Lower socioeconomic groups were less likely to report never or less than weekly consumption compared to daily consumption (*RRR* 0.67, *p*<0.05, CI 0.63–0.70). Similarly, they were less likely to report weekly consumption compared to daily consumption (*RRR* 0.83, *p*<0.05, CI 0.78–0.87). High socioeconomic groups were more likely to report never or less than weekly consumption of EDs over time compared to low SES (80% vs 75%); see S5 Table.

**School year.** For school year, Year 10s were the least likely to report never or less than weekly ED consumption (*RRR* 0.58, *p*<0.05, CI 0.53–0.62), with Year 8s being the most likely (*RRR* 0.81, *p*<0.05, CI 0.75–0.87). For weekly consumption Year 8s were more likely to report ED consumption (Year 8: *RRR* 1.04, *p* = 0.43, CI 0.95–1.13 vs. Year 10: *RRR* 0.93, *p* = 0.09, CI 0.85–1.01), although differences were not significant. Over time, 4% of Year 7's consumed ED's daily, compared to 6% of Year 9's, and 6% of Year 11's (see S6, S8 and S10 Tables).

## Demographic patterning of SSB and ED consumption overtime

The association between SES and SSB consumption changed over time. Where models used the reference category of 'daily consumption', lower socioeconomic groups were less likely to

**Table 3. Multinomial regression of ED consumption and sociodemographic characteristics with daily use as the reference category (n = 135,712).**

| | RRR | Std. Err | *p* | Confidence Intervals | |
|---|---|---|---|---|---|
| | | | | Upper bound | Lower Bound |
| **Daily use** *(base outcome)* | | | | | |
| **Never or less than weekly use** | | | | | |
| **Gender** | | | | | |
| *Girl* | 1.85 | 0.05 | <0.001 | 1.77 | 1.95 |
| **School Year** | | | | | |
| *Year 8* | 0.81 | 0.03 | <0.001 | 0.75 | 0.87 |
| *Year 9* | 0.61 | 0.02 | <0.001 | 0.57 | 0.66 |
| *Year 10* | 0.58 | 0.02 | <0.001 | 0.53 | 0.62 |
| *Year 11* | 0.66 | 0.03 | <0.001 | 0.61 | 0.72 |
| **Socioeconomic Status** | | | | | |
| *Low* | 0.67 | 0.02 | <0.001 | 0.63 | 0.70 |
| **Year** | 1.00 | 0.01 | 0.99 | 0.98 | 1.02 |
| **Weekly use** | | | | | |
| **Gender** | | | | | |
| *Girl* | 1.00 | 0.03 | 0.91 | 0.94 | 1.05 |
| **School Year** | | | | | |
| *Year 8* | 1.04 | 0.05 | 0.43 | 0.95 | 1.13 |
| *Year 9* | 0.97 | 0.04 | 0.49 | 0.89 | 1.06 |
| *Year 10* | 0.93 | 0.04 | 0.09 | 0.85 | 1.01 |
| *Year 11* | 1.00 | 0.05 | 0.93 | 0.91 | 1.09 |
| **Socioeconomic Status** | | | | | |
| *Low* | 0.83 | 0.02 | <0.001 | 0.78 | 0.87 |
| **Year** | 0.89 | 0.01 | <0.001 | 0.87 | 0.91 |

respond 'never or less than weekly' in SSB consumption compared to higher socioeconomic groups (*RRR* 0.91, *p*<0.05, CI 0.88–0.95). Descriptive data (S4 and S5 Tables) indicate that in 2002, consumption was very similar for higher and lower SES groups (e.g. 37% daily consumption for both groups), and while consumption has fallen for both groups, it has done so fastest in higher SES groups, leading to increased inequality (i.e. in 2017, 21% of young people from poorer families report daily consumption vs 16% of those from more affluent families). Regression models show that whilst both groups increased their reports of 'never or less than weekly' SSB consumption, the rate of increase was slower for low socioeconomic groups, indicating greater movement toward non-consumption in more affluent groups (S1 Fig). Likewise, lower socioeconomic groups were less likely to consume SSBs weekly compared to high socioeconomic groups (*RRR* 0.92, *p*<0.05, CI 0.89–0.95). S2 Fig shows that the socioeconomic pattern has changed over time in this model. In 2002, lower socioeconomic groups were more likely to consume SSBs weekly, however by 2004 this reversed with higher socioeconomic groups being more likely to using them weekly (compared to daily). As a result, the gap between low and high socioeconomic groups has widened, with lower socioeconomic groups being more likely to consume SSB's daily compared to weekly. This trend has changed in most recent years, with weekly consumption decreasing since 2015 for both socioeconomic groups.

The absence of time varying effects via other characteristics such as gender, school year and SES suggests that individual characteristics of SSB consumers have remained relatively stable over time, with RRR's being 1.00–1.01; likewise, for EDs consumption, with wider confidence intervals observed. For interaction estimates see S12 Table.

## Discussion

Present findings provide a profile of national trends over the past two decades of self-reported SSB and ED consumption among adolescents in Wales. This is the first large study to examine such consumption rates over time and by multiple demographic characteristics.

### Overall consumption trends

Almost 80 years since the first calls for a reduction in SSBs [5, 15] our findings provide an encouraging outlook on trends in SSB consumption among adolescents. We found that consumption reports since 1998 indicate a positive shift for daily SSB consumption with approximately 40% fewer adolescents reporting daily consumption in 2017 compared to 2000. A noticeable upward trend was observed for the number reporting never or less than weekly SSB consumption. ED consumption was not measured prior to 2013, but showed small decreases over time with one in four young people using EDs at least weekly in 2013. Recent findings in Denmark also displayed a decrease in daily SSB consumption between 2002 and 2018, albeit lower prevalence rates were observed in 2018 among Danish adolescents at 6.4% [24]. Compared to other HBSC countries, daily SSB consumption rates are somewhat lower than Malta, Belgium and Bulgaria at 34–37% [25]. The overall ED consumption rates for Wales are comparable to findings among Canadian adolescents [33] while lower rates were previously reported among a Korean population (11.4%) [34]. Comparable weekly SSB consumption rates were recently reported among Australians aged 15 or older [35]. While the sampling of differing populations, data collection tools and analyses makes direct comparisons of prevalence difficult, present findings provide the first insights into consumption trends of young people in Wales.

A number of environmental and policy changes may have contributed to the observed reduction in daily SSB consumption between 2000 and 2009. In 1996, the UK was reported to have one of the highest proportions of overall food advertisements worldwide, with 79% of adverts devoted to sweet or high fat foods. Since, more stringent advertising guidelines have been introduced to reduce young people's exposure to advertising of unhealthy food products. Between 2001 and 2007, a series of school food policies were also introduced across Wales in an attempt to improve the nutritional standards of food and drink provided in schools [36, 37]. Despite revealing a substantial decrease in daily SSB consumption rates, 52% of adolescents continue to consume SSBs on a weekly basis and 6% still consume EDs daily. As such, further political action is required to maintain downward trajectories, notwithstanding any impacts which may have since occurred because of the 2018 SIDL. Furthermore, as the global ED market is forecast to reach a net worth in excess of $84 billion by 2025 (projected 7% increase in sales) [12], it is of public importance that consumption trends among young people continue to be monitored.

### Socioeconomic patterning in consumption

We found clear patterning of SSB and ED consumption according to SES, observing higher consumption rates among young people from lower socioeconomic groups. These findings echo those of wider studies concerning adolescent SSB [25, 38, 39] and ED [38–40] consumption rates. With a lack of current legislation, the present findings are a potential reflection of the current marketing and availability landscape, with EDs being as affordable as SSBs [40] and some marketing trends disproportionately aimed at minority youth consumers [41]. A rapid UK-based review highlighted that 'own brand' EDs are often available at a cheaper price than water with young people preferring cheaper, less well-known varieties [42].

Our time trends analyses indicate that inequalities in SSB consumption have increased over time. While in 2002 there was no socioeconomic difference in SSB consumption, and declines

have been observed for both groups, these have been faster among children from more affluent families. Hence, whilst actions to date may have led to an improvement in SSB consumption at the population level, actions may have also inadvertently contributed towards growing inequality. A key rationale for introducing a SDIL in 2018 was the expected equitable impact on population health (observing greater health gains in those with the worst health problems). With earlier UK-based models projecting a potential widening of inequalities due to the SDIL however [43], unearthing the impacts of the SIDL among young people is vital as are urgent policies aimed at reducing inequalities.

Present findings are in line with the current obesity landscape, which also reveals a persistent widening of inequalities, as the gap between child obesity prevalence in the most and least deprived areas of Wales continues to broaden [44]. It is widely accepted that considerable effort will be required to halt the growing inequalities in obesity rates as child poverty is likely to increase and in turn inequalities will persist or worsen. As part of a nationwide strategy [20], the Welsh Government seeks to reduce the impact of ill health and inequality, which includes a reduction in the diet inequality gap between the most and least deprived communities. Present findings can inform the divergent trends noted among obesity in young people, yet at present, actions towards a UK-wide legislation for ED sales appear to have become stagnant [18] and the evaluation of the 2018 Sugar tax is ongoing [22].

Our findings have important implications for practitioners and policy makers alike, demonstrating how secular consumption trends are disparate between socioeconomic groups, an area which is pivotal for the introduction or modification of responsive interventions. Pinpointing the underlying factors which contribute to such socioeconomic differences is key to ensuring policy interventions facilitate healthy food choices for all population groups [23]. Future work will look to examine any differences in consumption rates across socioeconomic gradients in light of the introduction of the soft drinks levy in 2018.

**Limitations.** There are several limitations. First, despite the strength of utilising a large scale, national survey, findings are reliant upon self-reported data and therefore are subject to reporting bias. Second, data are derived from cross-sectional surveys and slight changes in survey questions have resulted in manipulation of data into categories. For example, socioeconomic data were not collected in 1998 and 2000 with SSB regression models estimated from 2002 and data include two extra measures of socioeconomic status from 2013 onwards. Third, the use of a binary measure of socioeconomic status may limit interpretations, despite being widely used. Fourth, we do not account for the variation in sample size which increases over-time. Fifth, as only three consumption categories are used, some detail is lost, but trends have been explored in S13–S16 Tables. Lastly, while SSBs and EDs are increasingly varied in terms of amounts of sugar and ingredients included, our single item measures treat these as homogeneous products. Our ability to comment on the content or volume of drinks consumed is limited, with a reliance on consumption frequency data only. This poses potential implications for the interpretation of our findings as despite a common perception that portion sizes have widely increased [45], UK trends in soft drink portion sizes over time remain unclear [46].

## Conclusion

Whilst overall reductions in SSB consumption are encouraging, study results indicate widespread continued consumption among adolescents and growing socioeconomic disparities in SSB consumption. There remains an urgent need for policy action to reduce adolescent consumption of SSBs, including EDs, and for these to be designed and evaluated with close attention to equity of impact.

## Supporting information

**S1 File. Information on missing data.**
(DOCX)

**S1 Table. Sample characteristics of the study participants (11–16 years) between 1998 and 2017 (n = 176,094).**
(DOCX)

**S2 Table. Boys and girls SSB consumption over-time.**
(DOCX)

**S3 Table. Boys and girls ED consumption over-time.**
(DOCX)

**S4 Table. High and low SES SSB consumption over-time.**
(DOCX)

**S5 Table. High and low SES ED consumption over-time.**
(DOCX)

**S6 Table. Year 7's SSB and ED consumption over-time.**
(DOCX)

**S7 Table. Year 8's SSB and ED consumption over-time.**
(DOCX)

**S8 Table. Year 9's SSB and ED consumption over-time.**
(DOCX)

**S9 Table. Year 10's SSB and ED consumption over-time.**
(DOCX)

**S10 Table. Year 11's SSB and ED consumption over-time.**
(DOCX)

**S11 Table. Cross-tabulation of sugary drink consumption and energy drink consumption.**
(DOCX)

**S12 Table. Interactions adjusted for other confounders; gender, school year and socioeconomic status (estimates in bold = $p < 0.05$).**
(DOCX)

**S13 Table. SSB over-time before recoding.**
(DOCX)

**S14 Table. SSB over-time after recoding.**
(DOCX)

**S15 Table. ED over-time before recoding.**
(DOCX)

**S16 Table. ED over-time after recoding.**
(DOCX)

**S1 Fig. Never or less SSB time trends according to socioeconomic grouping.**
(TIF)

**S2 Fig. Weekly SSB time trends according to socioeconomic grouping.**
(TIF)

## Acknowledgments

We would like to thank the young people and schools who took part in the study. We also would like to acknowledge Nicholas Page, who provided statistical advice regarding model interpretation.

## Author Contributions

**Conceptualization:** Kelly Morgan, Graham Moore.

**Data curation:** Chris Roberts, Simon Murphy.

**Formal analysis:** Emily Lowthian.

**Investigation:** Kelly Morgan, Emily Lowthian, Graham Moore.

**Methodology:** Emily Lowthian, Graham Moore.

**Project administration:** Kelly Morgan.

**Supervision:** Kelly Morgan, Graham Moore.

**Visualization:** Kelly Morgan, Emily Lowthian.

**Writing – original draft:** Kelly Morgan, Emily Lowthian, Jemma Hawkins, Britt Hallingberg.

**Writing – review & editing:** Kelly Morgan, Emily Lowthian, Jemma Hawkins, Britt Hallingberg, Manal Alhumud, Chris Roberts, Simon Murphy, Graham Moore.

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
