## [Decision Letter · Decision Letter 0]

11 Dec 2020

PONE-D-20-21401

Sugar-sweetened beverage consumption from 1998-2017: findings from the health behaviour in school-aged children/school health research network in Wales

PLOS ONE

Dear Dr. Morgan,

Thank you for submitting your manuscript to PLOS ONE. After careful consideration, we feel that it has merit but does not fully meet PLOS ONE’s publication criteria as it currently stands. Therefore, we invite you to submit a revised version of the manuscript that addresses the points raised during the review process.

We look forward to receiving your revised manuscript.

Kind regards,

Jane Anne Scott, PhD, MPH Grad Dip Dietetics, BSc

Academic Editor

PLOS ONE

Additional Editor Comments :

Lines 155 -160 Ethical approval  There is no reference to the SHRN surveys in this section

Reviewers' comments:

Reviewer's Responses to Questions

**Comments to the Author**

1. Is the manuscript technically sound, and do the data support the conclusions?

Reviewer #1: Partly

Reviewer #2: Yes

2. Has the statistical analysis been performed appropriately and rigorously? 

Reviewer #1: I Don't Know

Reviewer #2: I Don't Know

3. Have the authors made all data underlying the findings in their manuscript fully available?

Reviewer #1: Yes

Reviewer #2: Yes

4. Is the manuscript presented in an intelligible fashion and written in standard English?

Reviewer #1: Yes

Reviewer #2: Yes

5. Review Comments to the Author

Reviewer #1: Enjoyed reading this piece of work. Overall this is an interesting piece of work considering change of SSB and ED in adolescents using cross sectional data. There are a number of changes noted below for the authors to consider and a few questions to consider.

Abstract:

Can you add some figures to back up key findings? Is there an overall conclusion for paper re implications for PH?

Intro:

Line 60: Rather than ‘it is linked to’ can you make this sentence a little clear to say consumption of these drinks are linked to dental erosion and Type II diabetes

Line 62: Can you add an actual figure where you say ‘of which Eds make up a contribution’ as you have done for SSBs.

Line 68: Because you are considering both SSB and Eds can you also provide the context for SSBs re economic contribution.

Lines 100-102: The aims and objectives need to be made clearer and more specific for this paper. Make reference to the data you explored, the specific time periods, and also the key variables explored.

Overall comment: As you start with SSB discussion – can you restructure the introduction to discuss aspects related to SSBs then EDs to make it more succinct and flow better. You tend to focus more on EDs than SSBs in parts of intro so the rationale in lines 98-99 only capture the ED aspect as opposed to the importance of exploring the effect of SSB consumption in this age group as well. Also, is this analysis based on total daily consumption of SSBs and EDs?

Methods:

Lines 108-110: I appreciate you have referenced the data are appended as per previous studies and but can you provide more context? For example, how frequently are these surveys carried out between 1998-2017 – yearly, every 2 years etc?

Lines 107-110: I think the authors need to include more detail about the survey methods. In particular the method used to obtain dietary data (FFQs). This can then be linked to the section on ‘limitations’ where they mention the reliance on self-reported data.

Socio-demographics – this section includes more than that so may be better as participant characteristics as you have gender, year etc included

Line 116 and 121: FAS method, I’m not overly familiar with this method but can you explain why you only have two groups - low and high for SES effect? You should also note the limitations of this method for SES in the discussion. In this section you also mention SES was only available from 2002 – what did you use as SES categorisation for data pre-2002? Or was this excluded for these data? This needs noted. Did the change in SES coding have any impact (2002 and then in 2013 changed – if not tested that’s okay but then need to note in limitations categorisation of SES did change between these time points)

Line 129: Here you mention in the first two survey years for SSBs - can you provide the actual survey years and periods you have from 1998-2017 (I see this in S1 table but it would help the reader to be included in the text) and be specific that the ED data only includes 2013-2017; that is quite different (4y verse 20y)

Line 164: change to ‘using’ Stata15

Results:

Currently are lengthy. I’m not convinced all the supplementary tables are required as a lot are descriptive.

A few suggestions would be to condense the descriptive stats and focus more on the overall key findings from the more complex model results, once you have factored in year, SES and gender. Otherwise they get lost. Also you have significant interactions – can you provide more detail/explanation on the interaction effects rather than just refer to a supplementary table – this is one of the supplementary tables that may be better in the results findings.

Are you able to provide actual p-values as opposed to reporting as p<0.05; appreciate you have CI but if using the p-value too then would be better.

Re presentation of results: use more detailed subheadings, for example, as the effect of year on both SSBs and ED; and then another subheading noting the effect of gender, SES and year group and just highlight the key points.

Discussion

Can the authors provide some detail as to why they think there has been such a reduction in daily SSBs since 2002/9 – here there was a steep reduction in comparison to other survey periods, prior to the sugartax etc. Anything related to changes in school food and drink purchasing (relates back to a prior question about is this data total dietary data?) Switch from SSB to diet drinks?

Limitations:

• Need to acknowledge point about the SES coding, change from 2002,

• I wonder if the authors can comment on the robustness of the SES categorisation, in particular for this piece of research? I.e. possible changes in the ‘sensitivity’ of FAS over the time periods. In addition, the paper referenced (Currie et al, 2008) states “FAS associations are strong for health out-comes that are related to family culture and behaviour, but less so for some behaviours where peer norms area potentially powerful influence”. I wonder if the authors can comment on this in terms of the influence of peers in SSB consumption and the subsequent use of FAS to categorise SES?

• Acknowledge the SSB drinks aren’t categorised into different sugar contents, cross sectional survey – not individual-child changes.

• Acknowledgment of the limitations of FFQs (in adolescents) should be included.

Line 313: What does “other HBSC countries” mean? Is the same survey conducted in other countries? Just needs a brief explanation, or perhaps include more detail about the HBSC and SHRN surveys in the methods (as mentioned above).

Two other questions: does the model account for the variation in sample size? For example, in one of the later surveys roughly over 40,000 by comparison to much smaller survey numbers previously?

Reviewer #2: Thank you for this well-written manuscript. This research investigates trends in SSB consumption frequency among 11-16 year olds in Wales, via national school surveys, over a period of almost 20 years; and ED consumption over a 5-year period. Sociodemographic determinants of SSB and ED consumption frequency are explored. This research has a large sample size, and 9 data collection points over the time period. The work contributes a comprehensive report of the current and historical SSB consumption trends among adolescents in Wales, and highlights the socioeconomic inequality of this behaviour.

My main concern is around the grouping of response options. Around 50% of responses were categorised as ‘weekly’, defined as consuming SSB/ED between 1 and 6 times per week. There is a big difference between these frequencies. This should be discussed as a limitation, and the disaggregated response data provided as a supplementary table to assist with interpretation of findings. I have given some more specifics to this (points 5 and 7 below), along with some other minor recommendations (in order of appearance through the manuscript).

1. The aim should be clarified, to differentiate between the amount of SSB and ED data you have (ie 19 years vs 5 years), and the addition of the word ‘frequency’, to describe the consumption pattern that you are measuring.

2. In the methods, please give a brief overview of the study sampling strategies and procedures for each survey; including sampling methods, frequency of administration and year groups sampled at each time point. Were the same schools sampled every time, or new schools approached at each data collection wave? Also describe the approach that led to larger sample sizes at the 2015 and 2017 waves.

3. Please also indicate in the methods whether questionnaires were completed solely by the students, a parent proxy or a combination.

4. In the results (line 211) please revise the statement ‘remained stable at 6% each year’ – ED data were collected at 3 time points in the period, not each year.

5. Could the disaggregated SSB & ED response data be made available as a supplementary table and mentioned in your results? In particular, it would be useful to see the spread of intake responses that were grouped into the ‘weekly’ category at each time point.

6. The abstract and discussion both mention that the data is ‘nationally representative’, but this is not established in the methods or results. Please do so in at least one of those sections, and then add a comment to the discussion about the representativeness of the samples and generalisablty of your findings.

7. Please add a discussion point about whether these categorical groupings might lead to an overestimation of the scale of the decrease in SSB consumption. Moving from daily to weekly SSB consumption is a great outcome, however the true decline may be much more modest. For example, is it possible that this decline is the result of SSB intakes decreasing from 7 to 6 times per week? Is it also possible that this is reflected by the replacement of one of the SSBs per week with an ED? The recommended disaggregated data table (see point 5) might help you explore the likelihood of this limitation.

8. A further limitation to consider is the reporting of only frequency rather than incorporating the amount consumed at any occasion. As portion sizes of commercial foods appear to have increased over time, is it possible that total SSB consumption (eg in mls) has not declined?

6. PLOS authors have the option to publish the peer review history of their article (what does this mean?). If published, this will include your full peer review and any attached files.

Reviewer #1: No

Reviewer #2: **Yes: **Gemma Devenish

---

## [Author Response · Author response to Decision Letter 0]

25 Jan 2021

We would like to thank both reviewers for their time to review our manuscript and for their comments and suggested changes. Please find a point-by-point response document attached along with an updated manuscript.

---

## [Decision Letter · Decision Letter 1]

8 Mar 2021

Sugar-sweetened beverage consumption from 1998-2017: findings from the health behaviour in school-aged children/school health research network in Wales

PONE-D-20-21401R1

Dear Dr. Morgan,

We’re pleased to inform you that your manuscript has been judged scientifically suitable for publication and will be formally accepted for publication once it meets all outstanding technical requirements.

Kind regards,

Jane Anne Scott, PhD, MPH Grad Dip Dietetics, BSc

Academic Editor

PLOS ONE

Reviewers' comments:

Reviewer's Responses to Questions

**Comments to the Author**

1. If the authors have adequately addressed your comments raised in a previous round of review and you feel that this manuscript is now acceptable for publication, you may indicate that here to bypass the “Comments to the Author” section, enter your conflict of interest statement in the “Confidential to Editor” section, and submit your "Accept" recommendation.

Reviewer #1: All comments have been addressed

Reviewer #2: All comments have been addressed

2. Is the manuscript technically sound, and do the data support the conclusions?

Reviewer #1: Yes

Reviewer #2: (No Response)

3. Has the statistical analysis been performed appropriately and rigorously? 

Reviewer #1: Yes

Reviewer #2: (No Response)

4. Have the authors made all data underlying the findings in their manuscript fully available?

Reviewer #1: Yes

Reviewer #2: (No Response)

5. Is the manuscript presented in an intelligible fashion and written in standard English?

Reviewer #1: Yes

Reviewer #2: (No Response)

6. Review Comments to the Author

Reviewer #1: Many thanks to the authors for working through and responding to all the comments. These have all been considered and addressed, thank you. The responses and additions have helped clarify any initial queries. I have enjoyed reviewing and reading the topic of the paper and it is of PH importance.

Reviewer #2: (No Response)

7. PLOS authors have the option to publish the peer review history of their article (what does this mean?). If published, this will include your full peer review and any attached files.

Reviewer #1: No

Reviewer #2: **Yes: **Gemma Devenish

---

## [Editor Report · Acceptance letter]

22 Mar 2021

PONE-D-20-21401R1 

Sugar-sweetened beverage consumption from 1998-2017: findings from the health behaviour in school-aged children/school health research network in Wales 

Dear Dr. Morgan:

I'm pleased to inform you that your manuscript has been deemed suitable for publication in PLOS ONE. Congratulations! Your manuscript is now with our production department. 

Kind regards, 

on behalf of

Dr. Jane Anne Scott 

Academic Editor

PLOS ONE